# Assessment of Immune Responses to a Trivalent Pichinde Virus-Vectored Vaccine Expressing Hemagglutinin Genes from Three Co-Circulating Influenza A Virus Subtypes in Pigs

**DOI:** 10.3390/vaccines11121806

**Published:** 2023-12-02

**Authors:** Sushmita Kumari, Jayeshbhai Chaudhari, Qinfeng Huang, Phillip Gauger, Marcelo Nunes De Almeida, Hinh Ly, Yuying Liang, Hiep L. X. Vu

**Affiliations:** 1Nebraska Center for Virology, University of Nebraska-Lincoln, Lincoln, NE 68583, USA; skumari2@huskers.unl.edu (S.K.); jchaudhari2@huskers.unl.edu (J.C.); 2School of Veterinary Medicine and Biomedical Sciences, University of Nebraska-Lincoln, Lincoln, NE 68583, USA; 3Veterinary & Biomedical Sciences Department, College of Veterinary Medicine, University of Minnesota, Twin Cities, MN 55108, USA; huangq@umn.edu (Q.H.); hly@umn.edu (H.L.); 4Veterinary Diagnostic Laboratory, College of Veterinary Medicine, Iowa State University, Ames, IA 50011, USA; pcgauger@iastate.edu (P.G.); malmeida@iastate.edu (M.N.D.A.); 5Department of Animals Sciences, University of Nebraska-Lincoln, Lincoln, NE 68583, USA

**Keywords:** Pichinde virus, viral vector vaccine, swine influenza virus, vaccine platform

## Abstract

Pichinde virus (PICV) can infect several animal species and has been developed as a safe and effective vaccine vector. Our previous study showed that pigs vaccinated with a recombinant PICV-vectored vaccine expressing the hemagglutinin (HA) gene of an H3N2 influenza A virus of swine (IAV-S) developed virus-neutralizing antibodies and were protected against infection by the homologous H3N2 strain. The objective of the current study was to evaluate the immunogenicity and protective efficacy of a trivalent PICV-vectored vaccine expressing HA antigens from the three co-circulating IAV-S subtypes: H1N1, H1N2, and H3N2. Pigs immunized with the trivalent PICV vaccine developed virus-neutralizing (VN) and hemagglutination inhibition (HI) antibodies against all three matching IAV-S. Following challenge infection with the H1N1 strain, five of the six pigs vaccinated with the trivalent vaccine had no evidence of IAV-S RNA genomes in nasal swabs and bronchoalveolar lavage fluid, while all non-vaccinated control pigs showed high number of copies of IAV-S genomic RNA in these two types of samples. Overall, our results demonstrate that the trivalent PICV-vectored vaccine elicits antibody responses against the three targeted IAV-S strains and provides protection against homologous virus challenges in pigs. Therefore, PICV exhibits the potential to be explored as a viral vector for delivering multiple vaccine antigens in swine.

## 1. Introduction

Influenza A virus in swine (IAV-S) is a major respiratory pathogen affecting swine production worldwide [1]. Typically, IAV-S infection results in acute respiratory diseases with mild clinical signs and low mortality rates [2]. However, IAV-S infection significantly reduces infected pigs’ average daily weight gain and imposes a considerable economic burden on the swine industry [3,4]. Besides its economic impact, IAV-S poses a significant public health concern due to its zoonotic potential [5].

Three subtypes of IAV-S, H1N1, H1N2, and H3N2, are widespread in North America [6,7]. The H1 and H3 genes of IAV-S can be phylogenetically classified into lineages, which are further subdivided into clades. The percentage of pairwise nucleotide distances between the clades within a lineage can be as high as 16% [8,9,10]. The profound genetic diversity poses a significant challenge in developing broadly protective influenza vaccines (reviewed in [11,12,13]).

Whole-inactivated virus (WIV) vaccines are widely used to control IAV-S. These vaccines contain multiple IAV-S isolates representing the IAV-S linages cocirculating in the field. Generally, WIV vaccines confer robust protection against homologous but not against antigenically mismatched IAV-S strains [14,15]. Moreover, pigs vaccinated with WIV vaccines might exhibit more severe clinical diseases when exposed to antigenically mismatched IAV-S strains, a phenomenon known as vaccine-associated enhanced respiratory diseases (VAERD) [14]. Another limitation of WIV vaccines is that their efficacy is significantly reduced when administered to piglets with maternally derived antibodies (MDA) [16].

Live-attenuated influenza virus (LAV) vaccines are also commercially available. Unlike WIV vaccines, LAV vaccines confer partial protection against mismatched IAV-S isolates instead of inducing VAERD [17,18,19,20]. However, the LAV vaccine has the potential to revert to virulence after reassortment with field IAV-S [21].

Significant effort has been made to develop subunit vaccines targeting hemagglutinin (HA) protein, the surface glycoprotein responsible for viral attachment to the host cells. The HA protein-based vaccines elicit complete protection against homologous IAV-S strains [22]. However, the HA protein-based vaccine can induce VAERD, like WIV vaccines, when the vaccinated pigs are exposed to mismatched IAV-S strains [23].

DNA vaccines based on the HA gene have been tested in pigs. Immunization of pigs with a naked DNA plasmid results in weak immune responses, primarily due to ineffective cellular entry of the plasmid. Hence, various approaches, including in vivo electroporation and needle-free intradermal applicators, have been devised to enhance the immunogenicity of DNA plasmid vaccines [24,25,26,27,28,29]. Typically, hemagglutinin inhibition (HI) antibody titers are not detected in DNA-vaccinated pigs after one immunization dose. Moreover, pigs vaccinated with the DNA vaccines are not protected from developing severe lung lesions upon challenge infection with the homologous IAV-S strain. However, they may have shorter durations and lower magnitudes of virus shedding in their nasal cavity.

Self-replicating RNA replicon systems derived from positive-strand, single-stranded viral RNA genomes such as alphaviruses and pestiviruses have been utilized for delivering the HA genes to pigs [30,31,32]. In these systems, the structural genes of the alphaviruses or pestiviruses are replaced with genes encoding vaccine antigens, such as the HA gene of IAV-S. This way, the alphavirus or pestivirus nonstructural genes function as the RNA replication complex to amplify hybrid mRNA molecules. Pigs vaccinated with the self-replicating RNA replicon vaccines containing the IAV-S HA genes are protected against challenge infection with homologous IAV-S strains [31,32].

Several viral vectors, including replication-defective adenovirus, pseudorabies virus, and Orf virus have been utilized to deliver the HA gene [33,34,35,36,37,38,39,40]. In general, viral vector vaccines carrying the HA gene induce effective protective immune responses against the homologous IAV-S strain from which the HA gene was derived. For instance, intramuscular or intranasal vaccination of pigs with a replication-defective adenovirus type 5 (Ad5) encoding HA proteins has been shown to provide protective immunity against challenge infection from the homologous viral strains and partial protection against heterologous challenges [35,36,37,38].

Notably, pigs receiving the Ad5 viral vector or alphavirus RNA replicon vaccines do not develop VAERD after being challenged with a heterologous IAV-S strain [35]. These observations suggest that intracellular delivery of the HA antigen may help to prevent VAERD.

Pichinde virus (PICV), a nonpathogenic member of the *Arenaviridae* family, has a broad cell tropism and holds significant potential as a viral vector for vaccine advancement [41,42]. Recombinant PICV-vectored vaccines have been tested in mice, guinea pigs, and turkeys. In all cases, animals that received the PICV-vectored vaccines mounted robust immune responses against the vaccine antigens [41,42].

In a previous study, we demonstrated that a monovalent PICV-vectored vaccine expressing the HA gene of the H3N2 virus induced high titers of virus-neutralizing and hemagglutination inhibition antibodies and protected pigs against a subsequent challenge from the homologous H3N2 virus [43]. In the present study, we demonstrated that pigs vaccinated with a trivalent PICV expressing representative HA genes from three co-circulating IAV-S strains mounted robust immune responses against all three antigens, and were protected against challenge infection with the homologous H1N1 strain.

## 2. Materials and Methods

### 2.1. Cells, Reagents, and Viruses

Madin–Darby canine kidney (MDCK, ATCC CCL-34) and baby hamster kidney 21 (BHK-21, ATCC C-13) were obtained from the American type culture collection (ATCC, Manassas, VA, USA). MDCK cells were cultured in DMEM (Life Technologies, Grand Island, NY, USA) supplemented with 10% heat-inactivated fetal bovine serum (FBS, Sigma-Aldrich, St. Louis, MO, USA), 2.5% bovine serum albumin (BSA, Sigma-Aldrich, Saint Louis, MO, USA), and 25 mM HEPES (Hyclone, Life Technologies, Grand Island, NY, USA). BHK-21 cells were cultured in DMEM supplemented with 10% FBS. BSRT7 cells, which are BHK-21 cells stably expressing T7 RNA polymerase, were obtained from K.K. Conzelmann (Ludwig-Maximilians-Universität, Germany) and cultured in MEM supplemented with 10% FBS and 1 μg/mL Geneticin (Life Technologies, Carlsbad, CA, USA). 

Three IAV-S strains: A/swine/Iowa/A01202099/2011 (H1N1-pdm09), A/swine/Minnesota/A01392045/2013 (H1N2-δ1a), and A/swine/Texas/4199-2/1998 (H3N2-TX98) were obtained from the National Veterinary Services Laboratories (NVSL, Ames, IA, USA). Virus stock for all three IAV-S were grown in MDCK cells.

### 2.2. Vaccines Used in the Study

The whole-inactivated virus vaccine Flusure XP was purchased from Zoetis (Parsippany, NJ, USA). The vaccine was prepared and administered according to the label. The WIV contains four heat-killed IAV-S strains: A/Swine/Iowa/110600/2000 (H1N1), A/Swine/Oklahoma/0726H/2008 (H1N2), A/Swine/North Carolina/394/2012 (H3N2), and A/Swine/Minnesota/872/2012 (H3N2). 

The recombinant PICV-vectored vaccine was prepared as previously described [41]. The HA genes of three representative IAV-S strains, H1N1-pdm09, H1N2-δ1a, and H3N2-TX98, were codon-optimized for optimal protein expression in pigs and synthesized using a commercial DNA synthesis service (Genscript, Piscataway, NJ, USA). The HA gene from each of the three IAV-S strains was cloned into multiple cloning sites of both S1 and S2 plasmids and co-transfected into BSRT-7 cells along with a third plasmid encoding for L segment. Infectious recombinant PICV was rescued as previously described [41]. The resulting recombinant PICV modified to express the HA gene was plaque purified and amplified in BHK-21 cells. Virus titers were quantified using the conventional plaque assay in BHK-21 cells. The three PICVs expressing individual HA antigens were combined at a concentration of 10^6^ PFU/mL for each PICV to make a cocktail of the trivalent PICV-vectored vaccine (TPV) used in animal vaccination.

### 2.3. Vaccination and Challenge Experiment

The vaccination and challenge experiment were conducted in a weaned pig’s model, as described previously [43]. Twenty-one 4-week-old pigs, seronegative for IAV-S and PRRSV, were obtained from Midwest Swine Research. The pigs were randomly assigned into four treatment groups and were accommodated in the animal biosafety level 2 (ABSL2) research facility at the University of Nebraska-Lincoln (UNL). Group 1 consisted of three pigs that were neither vaccinated nor challenged with IAV-S (NV/NC). Groups 2, 3, and 4 consisted of six pigs each. Group 2 was injected intramuscularly (i.m.) with 2 mL of PBS to serve as a non-vaccination control. Group 3 was vaccinated i.m. with the WIV vaccine following the manufacturer’s instruction. Group 4 was inoculated i.m. with the TPV vaccine. The WIV and TPV vaccines were administered twice, with a three-week interval between injections.

Blood samples were collected from all the pigs on days 0, 21, and 42 post-vaccination (p.v.) and plasma from these samples was isolated and stored at −20 °C for later evaluation of the humoral immune response.

On day 42 post-vaccination (p.v.), pigs in groups 2 to 4 were challenged with 2 × 10^5^ TCID_50_ of the H1N1 pdm09 virus. The challenge virus was diluted in 4 mL of serum-free DMEM, with 2 mL administered via the intra-tracheal route and the remaining 2 mL given via the intra-nasal route, with 1 mL in each nostril. Pigs in group 1 were not subjected to the virus challenge and served as the no-vaccination/no-challenge (NV/NC) control.

Nasal swabs were collected daily post-challenge (p.c.) to measure potential IAV-S shedding. On day 5 p.c., all pigs were humanely euthanized. During necropsy, gross lung lesions were scored by a veterinary pathologist blinded to the treatment group. Bronchioalveolar lavage fluid (BALF) was collected in 50 mL of cold PBS to measure IAV-S RNA within the lungs. After that, lung tissue samples were collected and fixed in 10% buffered formalin to examine potential microscopic lesions.

### 2.4. Immunological Assay

Antibodies specific to the IAV-S nucleoprotein (NP) were detected using a commercial blocking ELISA (IDEXX, Westbrook, ME, USA). A sample-to-negative (S/N) ratio smaller than 0.6 indicated the presence of anti-NP antibodies in the plasma sample.

Antibody responses against H1N1-pdm09, H1N2-δ1a, and H3N2-TX98 viruses were measured using virus neutralization (VN) and hemagglutination inhibition (HI) assays [44]. The plasma samples were diluted 2-fold, serially after an initial dilution of 1:10. A titer of <1:10 was assigned the value of 5 for graphical representation purposes.

Virus neutralization against PICV was performed using plasma samples collected from TPV-vaccinated pigs on days 0, 21, and 42 p.v. The samples were diluted 2-fold, serially in DMEM in a 96-well plate and incubated with an equal volume containing 100 TCID_50_ of recombinant PICV expressing green fluorescent protein (PICV-GFP) for 1 h at 37 °C. After that, the plasma–virus mixture was transferred to a 96-well plate containing confluent BHK-21 cells seeded 24 h earlier. The plates were incubated for 48 h at 37 °C in a humidified atmosphere containing 5% CO_2_. The expression of GFP was visualized using a fluorescence microscope. Neutralization titers were defined as the reciprocal of the highest plasma dilution that showed complete inhibition of PICV-GFP infection.

### 2.5. Quantification of Viral Load

IAV-S RNA extraction and quantification from nasal swabs and BALF samples were performed as previously described [43]. Viral loads were reported as log_10_ copies of IAV RNA per 100 μL of sample used to extract RNA. For statistical purposes, samples with undetectable levels of IAV RNA were assigned a value of 0.8, equivalent to the assay detection limit.

### 2.6. Pathological Analysis of Lungs

Gross lung lesions were evaluated during necropsy by a veterinary pathologist who was blinded to the treatment groups. The percentage of lung consolidation was calculated as described previously [45]. Middle lung lobe sections were stained with hematoxylin and eosin (H & E), followed by a routine pathological procedure for histopathological evaluation [17].

To identify IAV-S infected cells within the tissue, in situ hybridization (ISH) targeting IAV-NP RNA was performed in sections of the middle lung lobe. Virus-infected cell frequency in the airway epithelium and pulmonary parenchyma was scored on a 5-point scale: 0 for no signals, 1 for minimal occasional signals, 2 for mild scattered signals, 3 for moderate scattered signals, and 4 for abundant signals.

### 2.7. Statistical Analysis

All statistical analyses were carried out using GraphPad Prism 9.3.1 Antibody titers were log_2_ transformed and analyzed using the mixed-effects analysis multiple comparisons. Gross lung lesion scores, lung microscopic lesion scores, and viral genome copies were analyzed using the Kruskal–Wallis test. Subsequently, the mean rank of each treatment group was compared with the mean rank of the PBS control group using the uncorrected Dunn’s test.

## 3. Results

### 3.1. Antibody Responses after Vaccination

All plasma samples that were collected before vaccination tested negative for IAV-S exposure with the commercial ELISA kit that detects antibodies specific to the IAV-S NP. As expected, pigs vaccinated with the TPV vaccine were negative for anti-NP antibodies throughout the 42-day post-vaccination period. On the other hand, five of the six pigs vaccinated with the WIV vaccine developed detectable levels of anti-NP antibodies by day 42 p.v. (Figure 1A).

Virus-neutralizing (VN) antibodies were measured against the three homologous IAV-S strains from which the HA genes were derived to generate the TPV vaccine. In the TPV group, VN antibodies against each of the three IAV-S subtypes were detected at day 21 p.v. and the neutralization titers increased after the booster immunization, reaching geometric mean titer between 1:533 and 1:1067 on day 42 p.v. (Figure 1B–D). Pigs in the WIV group developed VN antibodies against the three IAV-S strains. However, VN titers in the WIV group were only detected on day 42 p.v. As expected, pigs in the PBS and NV/NC groups did not exhibit VN titers against the three IAV-S strains tested (Figure 1B–D).

Next, we measured hemagglutination inhibition (HI) antibody titers against the three tested IAV-S strains. Notably, before vaccination, all pigs had a background HI titer of 1:40 against the H3N2 TX98 strain (Figure 1E) even though these samples tested negative for the presence of antibodies against the IAV-S NP (Figure 1A). On day 42 p.v., all six pigs in the TPV group had increased HI titers against the H3N2 TX98 virus. On the other hand, only one of the six pigs in the WIV group exhibited an increase in HI titers against the H3N2 TX98 virus, while none of the pigs in the NV/NC or PBS group showed any increase in HI titers against the H3N2 virus at day 42 p.v. (Figure 1E).

Unlike the case of H3N2, plasma samples collected before vaccination did not show any detectable HI titers against the H1N2-δ1a or H1N1-pdm09 strains (Figure 1F,G). On day 42 p.v., all pigs in the TPV group developed HI antibodies against the H1N1-pdm09 and H1N2-δ1a virus with a geometric mean titer of 1:45 and 1:71, respectively. For the WIV group, the mean HI titers of samples collected on day 42 p.v. against H1N1-pdm09 and H1N2-δ1a viruses were 1:45 and 1:10, respectively. Pigs in the NV/NC or PBS groups did not show HI antibody titers against the H1N2-δ1a and H1N1-pdm09 viruses at any sampling days (Figure 1F,G). 

### 3.2. Viral Genomic RNA in Nasal Secretions and Lungs

All pigs in the PBS group exhibited detectable levels of the IAV-S genomic RNA copies in their nasal swabs, starting from day 1 after challenge infection with the H1N1-pdm09. The IAV-S genomic RNA copies peaked on day 4 p.c., reaching the mean of 10^7.6^ copies per 100 μL of sample (Figure 2A). Five of the six pigs in the WIV group had detectable IAV-S genomic RNA copies in their nasal secretions starting from day 1 p.c, while only one pig in the TPV group had relatively low copies IAV-S genomic RNA on days 2 and 4 p.c. (Figure 2A).

The area under the curve (AUC) was calculated to assess the overall extent of IAV-S shedding across the 5-day observational period. The TPV group had a significantly smaller AUC than the PBS group (Figure 2B). The WIV group had a comparatively smaller AUC than the PBS group, but this difference did not reach statistical significance. As expected, no AUC values were observed in the NV/NC group, because no IAV-S genomic RNA was detected in nasal swabs of those pigs.

On day 5 p.c., BALF samples were collected to quantify the IAV-S RNA genome copies present within the lungs. All pigs in the PBS and WIV groups had high IAV-S genomic RNA copies in their BALF samples (Figure 2C). Conversely, only one pig in the TPV group had detectable IAV-S genomic RNA in its BALF sample. No IAV-S genomic RNA was detected in the BALF samples collected from pigs in the NV/NC group (Figure 2C).

### 3.3. Macroscopic and Microscopic Lung Lesions

No pigs in the NV/NC group showed lung consolidation indicative of IAV-S infection (Figure 3A). Conversely, all pigs in the PBS group displayed lung consolidation, varying from 0.5% to 4.5% (Figure 3A,B). Only one pig from the TPV group and two from the WIV group exhibited lung consolidation. Both TPV and WIV groups demonstrated significantly lower levels of lung consolidation compared to the PBS group (Figure 3B).

Pigs in the NV/NC group did not exhibit noticeable microscopic lung lesions (Figure 3C,D). The lungs from the pigs in the PBS group presented mild peribronchiolar lymphocytic cuffing, necrosis of bronchiolar epithelial lining, and low levels of interstitial pneumonia (Figure 3C), with the composite microscopic lung scores ranging between 0 and 9 (Figure 3D). Three out of the six pigs in the WIV group displayed microscopic lesions, scoring between 1 and 3. Only one pig in the TPV group showed mild microscopic lesions. Consequently, the TPV group’s microscopic lung lesion scores were statistically lower than those of the PBS group (Figure 3C,D).

An in-situ hybridization (ISH) assay was utilized to detect IAV-S infected cells within the lung tissue. The lung samples from pigs in the PBS group displayed a high number of virus-infected cells, particularly within the bronchial and bronchiolar epithelium (Figure 3E). Conversely, virus-infected cells were rarely detected in the lung tissues of pigs in the TPV or WIV group. As a result, both the TPV and WIV groups exhibited significantly lower ISH scores compared to the PBS group (Figure 3F).

### 3.4. Antibody Responses against the PICV Vector (Anti-Vectored Immunity)

Antibody responses against the viral vector could potentially hamper the vaccine’s effectiveness. Therefore, we performed a virus-neutralization assay to detect PICV-specific neutralizing antibodies. For the virus-neutralization assay, we employed PICV-GFP, enabling real-time monitoring of the virus-infected cells. Plasma samples obtained from pigs vaccinated with TPV on days 0 and 21 p.v. did not display any inhibition of PICV-GFP infection in BHK-21 cells. This was evident from the comparable numbers of GFP-positive cells observed in wells treated with these samples and in control cells without any plasma. However, at a dilution of 1:4, samples collected on day 42 p.v. showed some neutralization effects against the virus but were insufficient to prevent infection (Figure 4) completely. Together, the data demonstrated that pigs in the TPV group did not induce significant neutralizing antibodies against the viral vector. 

## 4. Discussion

Three IAV-S subtypes—H1N1, H1N2, and H3N2—are co-circulating in U.S. swine herds [6]. Due to the substantial genetic and antigenic differences, viruses within these three subtypes do not confer cross-protection against each other. Thus, we aimed to develop a trivalent PICV-vectored vaccine containing representative HA genes of these three IAV-S subtypes to broaden the vaccine antigenic coverage. However, antigenic competition might occur when multiple vaccine antigens are combined within a vaccine. Thus, this study is primarily centered on evaluating the antibody responses in pigs immunized with the trivalent PICV vectored vaccine (TPV). Our results demonstrate that pigs receiving the TPV exhibited VN and HI antibodies at similar titers against each of the three IAV-S strains, from which the HA genes were utilized in developing the TPV vaccine. Furthermore, pigs vaccinated with the TPV vaccine were protected against challenge with the H1N1-pdm09 virus. Together, these results indicate that the trivalent PICV vaccine elicits a strong and non-interfering immunity against all three IAV-S subtypes.

In this study, we included a group of pigs vaccinated with the WIV for reference purposes. However, it is essential to note that a direct comparison of immune responses and protection outcomes between pigs vaccinated with the TPV vaccine and those vaccinated with the WIV vaccine is challenging, primarily due to differences in the antigenic composition of these two vaccines. The TPV vaccine contains HA genes that match the three IAV-S strains employed in the HI and VN assays. In contrast, the WIV vaccine comprises four IAV-S strains genetically different from the three strains used in the VN and HI assays. Notably, the WIV vaccine lacks the H1N1-pdm09 strain, which explains the absence of detectable HI titers against this virus strain in WIV-vaccinated pigs. Following challenge infection with the H1N1-pdm09 virus, WIV-vaccinated pigs exhibited no statistically significant differences in viral loads in nasal swabs or BALF samples collected during necropsy compared to the PBS control group. Intriguingly, WIV-vaccinated pigs displayed less severe gross lesions than the PBS control group. Furthermore, very few IAV-S-infected cells were detected in lung sections of the WIV-vaccinated pigs. It is plausible that other IAV-S strains present in the WIV may induce partial cross-protective immunity against the H1N1-pdm09 strain.

Pigs inoculated with the WIV vaccine displayed detectable antibodies against the IAV-S nucleoprotein, whereas those administered with the TPV vaccine did not exhibit such antibodies. Consequently, it is possible to serologically differentiate TPV-vaccinated pigs from those naturally infected with IAV-S by employing a commercially available ELISA designed for detecting antibodies against the IAV-S NP. This serological distinction is important for disease control and eradication programs, as it enables the detection of naturally infected animals within the vaccinated population. This, in turn, allows for the implementation of appropriate measures, such as quarantine or culling of the infected herds, to effectively eliminate the viruses from the animal herd [46,47,48].

In this study, all pre-vaccination samples consistently displayed a baseline HI titer of 1:40 against the H3N2 virus. For pigs in the PBS or NV/NC group, a baseline HI titer of 1:40 against the H3N2 virus was observed throughout the study period. However, samples collected from pre-vaccination, or the PBS and NV/NC groups, did not exhibit antibodies against the IAV-S nucleoprotein, as determined with a commonly used commercial blocking ELISA for IAV-S serodiagnosis. Additionally, these samples did not show VN antibody titers against the H3N2 TX98 strain. We therefore concluded that the pigs used in this study had not been previously exposed to IAV-S, nor had they received maternally derived anti-IAV-S antibodies. We believe that the HI titers against the H3N2 virus in the pre-vaccination samples were likely the result of nonspecific inhibition. It is notable that the background HI titer was only observed against the H3N2 virus but not against the H1N1 or H1N2 viruses. We observed a similar pattern in a previous study, in which all pre-vaccination samples also displayed a baseline HI titer of 1:40 against the H3N2 TX98 strain [43]. Before performing the HI assay, all samples were treated with the receptor-destroying enzyme and pre-adsorbed to red blood cells to eliminate potential nonspecific inhibitors. Thus, the underlying reasons for the background HI titers against H3N2 TX98 observed in this study remain unknown.

Antibody responses directed against the viral vector can present a formidable obstacle in the development of viral vector vaccines. In the present study, no neutralizing antibodies against PICV were detected in samples collected on day 21 p.v. Samples collected after the booster immunization reduced PICV infection only at the lowest plasma dilution (1:4), indicating that the pigs did not mount a significant anti-vector antibody response. This finding is further supported by the observation that both VN and HI antibody titers against the three IAV-S strains significantly increased after the booster immunization. A prior study also demonstrated that mice and guinea pigs, when vaccinated with a PICV-vector vaccine expressing influenza virus antigens, exhibited significantly elevated antibody titers against the influenza virus upon receiving repeated immunizations [41]. Consequently, the absence of neutralizing antibody responses against the PICV vector makes it suitable for repeated immunization protocols.

## 5. Conclusions

We developed a trivalent PICV-vectored vaccine that contains HA genes from three concurrently circulating IAV-S subtypes in the U.S. This trivalent vaccine effectively elicited VN and HI antibodies against all three representative IAV-S strains. Notably, pigs immunized with the trivalent PICV-vectored vaccine were fully protected against challenge infection with the H1N1-pdm09 strain, one of the three IAV-S strains whose HA gene was included in the PICV-vectored vaccine. Further studies are required to assess the protective efficacy of the TPV vaccine against other IAV-S strains.

## Figures and Tables

**Figure 1 vaccines-11-01806-f001:**
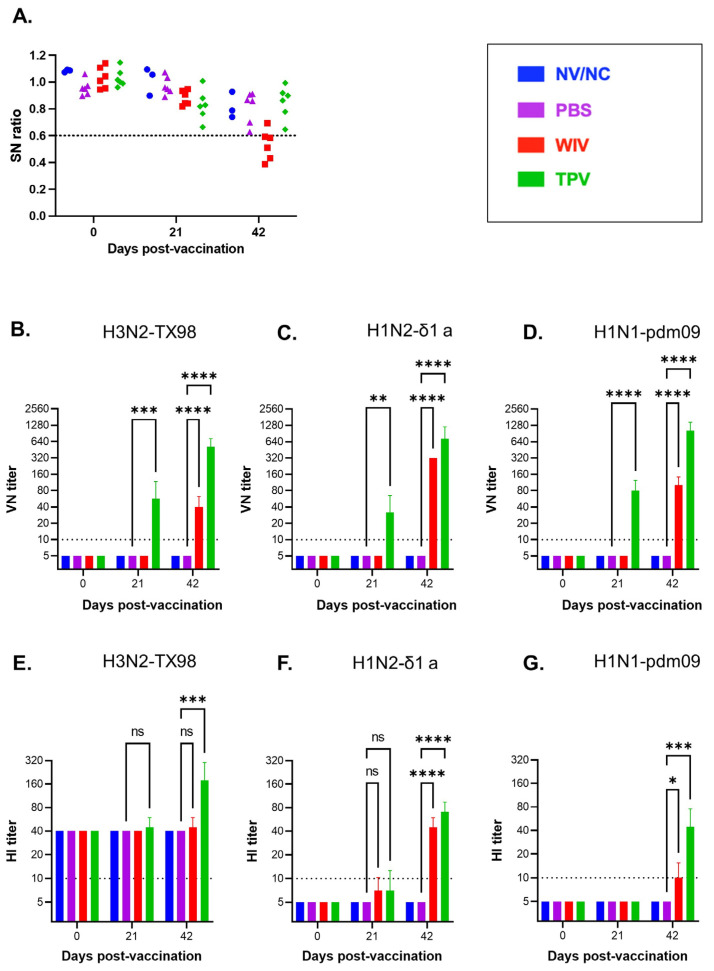
**Antibody response.** (**A**) Antibodies against IAV-S NP. Data are expressed as the sample-to-negative (S/N) ratio. A dotted line at S/N of 0.6 represents the cut-off limit for the assay. Samples with S/N value above the cutoff line are considered negative for the NP antibodies. NV/NC, neither vaccinated nor challenged. PBS, mock vaccination. WIV, whole-inactivated virus vaccine. TPV, trivalent PICV-vectored vaccine. (**B**–**D**) Virus-neutralizing antibody titers against the H3N2-TX98, H1N2-δ 1a, and H1N1-pdm09 strains, respectively. (**E**–**G**) Hemagglutination inhibition (HI) antibody titers measured against the H3N2-TX98, H1N2-δ 1a, and H1N1-pdm09 strains, respectively. For the VN and HI, the dotted lines at the dilution of 1:10 indicate the assay’s detection limit. Samples with undetectable activity at this dilution were considered negative and were assigned a value of 5 for graphical and statistical purposes. ns: no significant, * *p* ≤ 0.05; ** *p* ≤ 0.01; *** *p* ≤ 0.001; **** *p* ≤ 0.0001.

**Figure 2 vaccines-11-01806-f002:**
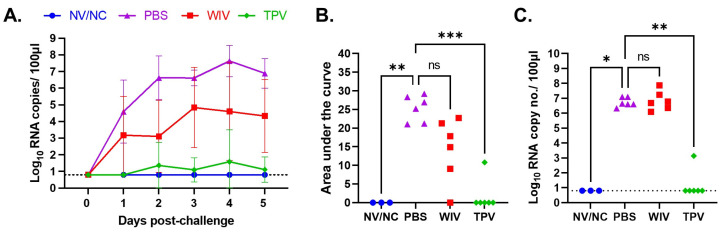
**Viral genomic RNA shedding in pigs post-challenge with H1N1-pdm09 virus.** (**A**) IAV-S genomic RNA copies in nasal swabs. (**B**) The area under the curve of viral loads in the nasal swabs. (**C**) IAV-S genomic RNA copies in bronchoalveolar lavage fluid (BALF) collected on day 5 p.c. The dotted horizontal lines at y = 0.8 log_10_ represent the assay detection limit. ns: non-significant, * *p* ≤ 0.05; ** *p* ≤ 0.01; *** *p* ≤ 0.001.

**Figure 3 vaccines-11-01806-f003:**
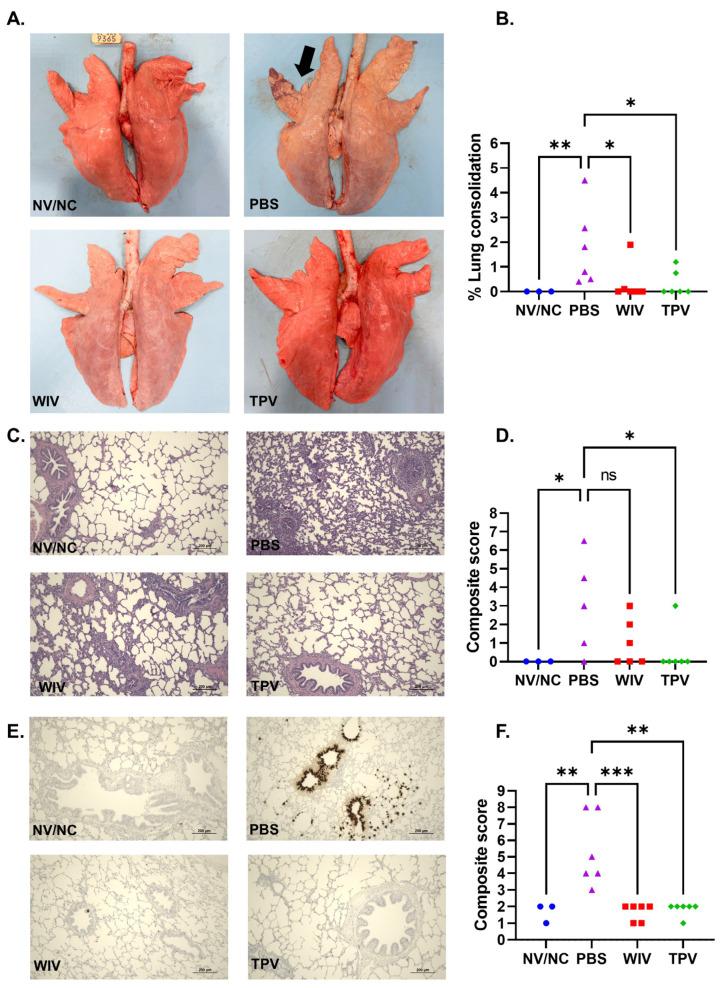
**Pathological assessment of pig lungs after challenge infection with the H1N1-pdm09 virus.** (**A**) Pictures of lungs from four treatment groups taken during necropsy. The black arrow indicates typical lesions associated with IAV-S infection. (**B**) Percentage of lung consolidation calculated based on the weighted proportions of each lobe to the total lung volume. (**C**) Images of lung section stained with H & E. (**D**) Composite score of microscopic lung lesion. (**E**) Images of lung sections stained with ISH to detect the IAV-S NP mRNA transcripts. (**F**) Composite scores of the lung section stained with ISH. ns: non-significant, * *p* ≤ 0.05; ** *p* ≤ 0.01; *** *p* ≤ 0.001.

**Figure 4 vaccines-11-01806-f004:**
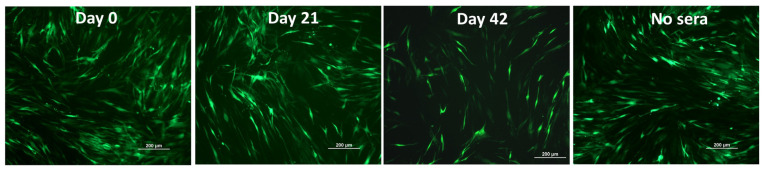
**Assessment of neutralizing antibodies against the PICV vector.** Plasma samples obtained from pigs in the TPV group on days 0, 21, and 42 p.v. were subjected to a 2-fold serial dilution in DMEM in a 96-well plate and then incubated with an equal volume of 100 TCID_50_ of the PICV-GFP. After a 1 h incubation, the plasma–virus mixture was transferred to a new 96-well plate containing BHK-21 cells. GFP signals were visualized using a fluorescence microscope at 48 h post-inoculation. The displayed images correspond to the plasma samples at the dilution 1:4.

## Data Availability

All relevant data are presented in the article.

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
