# Peer review of "Assessment of Immune Responses to a Trivalent Pichinde Virus-Vectored Vaccine Expressing Hemagglutinin Genes from Three Co-Circulating Influenza A Virus Subtypes in Pigs"

_vaccines, 2023, doi:10.3390/vaccines11121806_

Round 1

Reviewer 1 Report

Comments and Suggestions for Authors

1. The vaccine is targeted at three subtypes of IAV-S. Why only use H1N1 strains to attack the vaccine, but not H1N2 and H3N2 strains when judging the efficacy of the vaccine?

2. Please explain why the antibody against the vector did not increase significantly after two doses of immunization?

3. The method of detecting H3N2 antibodies is not accurate. What should be done?

4. Are the two words "100 L" in Part 3.2 incorrect?

5. There is no H1N1 in the WIV vaccine, so why set this trial group?

Author Response

We thank the reviewer for the careful evaluation of our manuscript. Our point-by-point response to the reviewer's comments is below. 

Comment 1: The vaccine is targeted at three subtypes of IAV-S. Why only use H1N1 strains to attack the vaccine, but not H1N2 and H3N2 strains when judging the efficacy of the vaccine?

Response: We concur with the reviewer's suggestion of conducting three distinct pig experiments to assess the vaccine's efficacy against all three IAV-S strains. However, financial constraints prevented us from pursuing this comprehensive approach. In the realm of influenza viruses, virus-neutralizing (VN) and hemagglutination inhibition (HI) antibodies serve as reliable indicators of vaccine-induced protection. Our study successfully demonstrated that the trivalent Pichinde virus vectored vaccine elicited VN and HI antibodies against the three IAV-S strains. Moreover, pigs vaccinated with this vector showed protection against the H1N1-pdm09 virus upon challenge. Given these outcomes, we anticipate that the trivalent vaccine should confer protection against the remaining two IAV-S strains. Notably, our prior study showcased the monovalent PICV vaccine, expressing the H3N2 HA gene, providing complete protection against the H3N2 virus. We hope these promising results will help us secure additional funding for additional studies to thoroughly evaluate the safety and efficacy of the PICV vector vaccine in pigs.

Comment 2: Please explain why the antibody against the vector did not increase significantly after two doses of immunization?

Response: In this study, we evaluated the neutralizing antibody, not the binding IgGs, against the vector (PICV). Pichinde virus (vector) glycoprotein has one of the highest numbers of N-linked glycans, allowing the virus to effectively evade neutralizing antibodies due to glycan shield (PMID 26587982). The lack of a detectable level of anti-vector neutralizing antibody even after multiple doses of immunization is a major strength of this Pichinde viral vector.

Comment 3: The method of detecting H3N2 antibodies is not accurate. What should be done?

Response: The virus-neutralization (VN) and hemagglutination inhibition (HI) assays utilized for detecting H3N2 antibodies are well-established and are standard procedures in our laboratory. Therefore, we have confidence in the accuracy of these tests. As we discussed in the manuscript (Lines 527-543), the samples did not exhibit virus-neutralization activities. Similarly, they all tested negative for antibodies against IAV-S nucleoprotein. Thus, we believe that these pigs were not infected with the influenza virus before they were included in this study.

Comment 4: Are the two words "100 L" in Part 3.2 incorrect?

Response: We could not identify the words “100 L” as the reviewer indicated. We assume the reviewer referred to 100 ul (microliter) in Part 3.2, and have confirmed its accuracy.

Comment 5: There is no H1N1 in the WIV vaccine, so why set this trial group?

Response: The WIV utilized in this study (FluSureXP) is a commercially available and widely employed multivalent vaccine within the field. It encompasses various IAV-S strains, including the H1N1 influenza virus A/Swine/Iowa/110600/2000. Consequently, we opted to incorporate it into our study for reference purposes. Acknowledging that the virus strains present in the WIV vaccine differ antigenically from those used in our trivalent vaccine, we explicitly articulated in the Discussion section that conducting a direct comparison between these two vaccine types would be unjustified.

Reviewer 2 Report

Comments and Suggestions for Authors

In this manuscript, the authors evaluated the immunology responses of using the recombinant tri-segmented Pichinde virus (rPICV) as a viral vector to deliver trivalent HA antigens in pigs. At the same time, they verified the protection of this vaccine against homologous virus challenge. I believe the current study is a big step up from their previous work. I have the following suggestions that might speed up the publication of this research.

1.     Please add the schematic views of reverse genetics system and study schedule in methods section.

2.     Line 130: The virus inoculation should be indicated in pfu per pig.

3.     Line130/154/157/298: TCID50 and CO2 should be unified, respectively.

4.     Line 220: The results of “HI titers against the H1N1-pdm09 or H1N2-δ1a strains” should cite Fig 1F and Fig 1G, not FIG. 1C and 1D.

5.     Line 209/252/268: No significance generally means p > 0.05. Please correct all annotations in figure legends.

6.     Line 257: Fig 4b should be 3B.

7.     The trivalent Pichinde virus-vectored vaccine expressing HA genes of IAVs showed expected immune responses and protective efficacy against H1N1 pdm09 virus. I recommend to add the results of cross-protection effects against various IAVs challenge or propose a related study plan for the future work in discussion section.

Author Response

We thank the reviewer for the careful evaluation of our manuscript. 

Comment 1: Please add the schematic views of reverse genetics system and study schedule in methods section.

Response: We appreciate the reviewer’s suggestion. Yet, the schematics detailing the reverse genetics system and the animal study have been presented in our prior publications (PMID: 26676795, 36146478). As per the publisher's guidelines, reiteration of these schematics is discouraged. Consequently, we did not include them in the current manuscript.

Comment 2: Line 130: The virus inoculation should be indicated in pfu per pig.

Response: Both TCID50 and pfu are widely adopted methods for titrating viruses. In our laboratory, the TCID50 method has been our primary choice for titrating influenza viruses. Hence, the influenza virus inoculation was administered at TCID50 per pig. Conversely, our collaborators, Drs. Liang and Ly, typically employ the pfu method to titrate their PICV vector. Consequently, the PICV vaccine dose was expressed as pfu per pig.

Comment 3:  Line130/154/157/298: TCID50 and CO2 should be unified, respectively.

Response: This is corrected in the text

Comment 4: Line 220: The results of “HI titers against the H1N1-pdm09 or H1N2-δ1a strains” should cite Fig 1F and Fig 1G, not FIG. 1C and 1D.

Response: This is corrected in the text

Comment 5:  Line 209/252/268: No significance generally means p > 0.05. Please correct all annotations in figure legends.

Response: This is corrected in the text

Comment 6:  Line 257: Fig 4b should be 3B.

Response: This is corrected in the text

Comment 7: The trivalent Pichinde virus-vectored vaccine expressing HA genes of IAVs showed expected immune responses and protective efficacy against H1N1 pdm09 virus. I recommend to add the results of cross-protection effects against various IAVs challenge or propose a related study plan for the future work in the discussion section.

Response: We agree with the reviewer that it would be ideal to conduct three separate pig experiments to evaluate the vaccine efficacy against all three IAV-S strains. However, the expenses associated with the pig experiment hindered our ability to proceed with this approach. a sentence proposing additional studies, as per the reviewer's suggestion.

Reviewer 3 Report

Comments and Suggestions for Authors

The authors of this paper demonstrated the efficacy of Pichinde virus (PICV) expressing the HA antigens of H3N2, H1N2, and H3N2 influenza A virus of swine (IAV-S) as a trivalent vector vaccine in pigs. Pigs vaccinated with this trivalent vector vaccine induced neutralizing and also HI antibodies against each IAV-S subtype of virus. Furthermore, it was shown to inhibit post-challenge infection with the H1N1 strain. These results strongly demonstrate the effectiveness of the trivalent vector vaccine against IAV-S created by the authors. More interestingly, neutralizing antibodies against PICV, which was used as a viral vector, were not induced in pigs vaccinated with the trivalent vector vaccine. This result seems to suggest that PICV is promising as a viral vector for vaccination. This is a very interesting paper and interesting for readers.

[Major point]

Nothing

[Minor point]

1. Line 130: “105” should be “105”. “5” should be superscript!

2. Line 131: “TCID50” should be “TCID50”. “50” should be subscript!

3. Line 157: “CO2” should be “CO2”. “2” should be subscript!

4. Lines 188, 220, 226...: “(FIG.   )” should be “(Figure   )”. The expression is different depending on the location in this paper, so it will be unified.

Author Response

We thank the reviewer for the careful review and the overall positive evaluation of our manuscript. 

Minor comment 1: Line 130: “105” should be “105”. “5” should be superscript!

Minor comment 2: Line 131: “TCID50” should be “TCID50”. “50” should be subscript!

Minor comment 3: Line 157: “CO2” should be “CO2”. “2” should be subscript!

Minor comment 4: Lines 188, 220, 226...: “(FIG.   )” should be “(Figure   )”. The expression is different depending on the location in this paper, so it will be unified.

Response: Thank you so much for your suggestions, the corrections have been implemented.